# The Reptile Relocation Industry in Australia: Perspectives from Operators

**Chantelle M. Derez * and Richard A. Fuller** 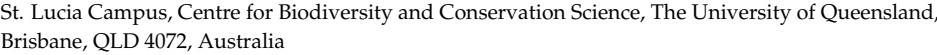

St. Lucia Campus, Centre for Biodiversity and Conservation Science, The University of Queensland, Brisbane, QLD 4072, Australia

* Correspondence: chantelle.derez@uq.net.au

**Abstract:** Thousands of reptiles are relocated annually in Australia, yet there has been relatively little research aimed at understanding how the reptile relocation industry operates. An online questionnaire was distributed to anyone who had relocated a reptile between April 2019 and April 2020, including wildlife relocators, wildlife rehabilitators and the general public. The questionnaire explored demographics, decision-making and concerns about how the industry functions, through 24 questions and two opportunities to provide open-ended comments. We received 125 responses and 123 comments from operators in all Australian states and territories. Beliefs about appropriate times and places for reptile releases were not reflected in practice for the majority of operators. Confidence about reptiles remaining at recipient sites was low regardless of how many years' experience an operator had. Escaped captive native reptiles were encountered by most operators, and a quarter of operators were called out to exotic non-native snakes. Operators across all levels of experience indicated a need for changes within the industry, including increased training and professionalism, and more scientific studies on the outcomes of relocations to address concerns about the impacts that the industry has on the wildlife that it is trying to protect.

**Keywords:** relocation; translocation; nuisance wildlife; human–wildlife conflict

## 1. Introduction

A substantial global industry has formed around removing nuisance or problematic wildlife from areas of human habitation, although the success of such translocations is poorly documented [1,2]. Relocated animals range from those posing a threat to human safety, often large carnivores (for example, leopards *Panthera pardus fusca* [3]; American black bears *Ursus americanus* [4], saltwater crocodiles *Crocodylus porosus* [5] and large herbivores (e.g., African elephants *Loxodonta africana* [6])), to 'nuisance' wildlife creating property damage or public annoyance [7,8]. Nuisance wildlife are often abundant species that are not of immediate conservation concern—for example, racoons *Procyon lotor* [9], Canada geese *Branta canadensis* [10], Australian white ibis *Threskiornis molucca* [11] and long-tailed macaques *Macaca fascicularis* [12]. Often, the perceived conflict is greater than the actual problem [13]. This creates a heightened awareness of a species' presence, which is then considered undesirable in the immediate environment of human residents, regardless of the actual level of threat [14,15]. A common response to nuisance wildlife is to exterminate the offending animal or remove it [16–18]. The purpose of relocating nuisance wildlife is not to create self-sustaining populations as in conservation-driven translocations, nor population survival as in mitigation translocations. Instead, often, the purpose is to eliminate the immediate threat or source of discomfort to the humans involved, resulting in animal welfare being considered of only secondary importance [19]. Understanding how decisions are made around nuisance wildlife relocations is a crucial first step in determining whether the outcomes of such efforts can be improved, and this forms a key focus of this paper.

One difficulty in managing these sorts of human–wildlife conflicts is defining nuisance species in the first place; what may be an annoyance to one person may not disturb another

person at all [20,21]. For example, feeding wildlife is a common pastime in many countries, with millions of dollars spent annually on bird seed [22]. The feeder gains enjoyment from attracting wildlife, although neighbours may not perceive the same positive benefits. Complaints may be made about noise, damage to gardens, smells, faeces, causing disturbances to domestic animals and reducing fear in wild predatory animals [7,20,23]. Additionally, food or scraps may attract other, less desirable species (e.g., rats, mice, squirrels), which then can attract predators (e.g., raccoons, monkeys, bears [24–27]). Ultimately, convenience and differing human attitudes may play a major role in the extent to which nuisance wildlife relocation occurs in a neighbourhood [3].

Ethical criteria, such as those set by the International Union for Conservation of Nature (IUCN) for conservation-based translocations may not be considered (or even known by) commercial or small-scale nuisance wildlife operators, and are also often absent from industry regulation, highlighting a divide between welfare principles and the practice of nuisance wildlife relocations [19]. Moreover, very rarely is the fate of released animals monitored, and thus it is difficult for the operator to receive feedback on the consequences of the decisions that they make during the capture and release of the animal [19,28].

Reptiles, especially snakes, provide a useful case study of the issues surrounding the relocation of nuisance wildlife [29,30]. Snakes have an unpopular reputation, invoking a deep-seated fear among many people for their personal safety and the safety of loved ones and domestic pets [31–33]. Snakes are frequently considered dangerous or undesirable in human environments, and thus present a human–wildlife conflict at the intersection between what is considered directly threatening or merely unwanted or reviled. Snakebite has been listed as a global health threat, and recent studies in urban areas have focused on identifying species of medically dangerous reptiles that are frequently encountered by people [32,34,35]. In socioeconomically disadvantaged areas, non-profit organisations often exist to deal with human–reptile conflict, such as Snakes of Namibia, who provide awareness, research and conflict mitigation [32]. Over a three-year period, 509 snakes were relocated in Windhoek, the capital of Namibia, by Snakes of Namibia [32]. Governmental responders are rare, but examples include Brazil's 'Environmental Police', a sub-division of the Military Police that deals with human–wildlife conflicts such as the illegal trade of wildlife and nuisance wildlife relocations [34]. The Rapid Response team from Kannur Forest Department in India responded to 1427 snake removal callouts over a three-year period [36]. In affluent areas, businesses have emerged that provide a commercial reptile removal service [31]. The government of the Northern Territory, Australia, responds to requests for nuisance wildlife removal during office hours, and forwards additional inquiries to private companies [35]. Over six years, the number of callouts for snake relocations in Darwin ranged between 631 and 851 per year [35]. In 2004, an average of 1232 snakes were requested to be relocated among 14 operators in Victoria, Australia, and 231 call-outs by seven operators and 'hundreds' of call-outs by another operator in Eastern Australia were received during 2006 [37,38].

Decisions about where to release these nuisance animals are up to the operator and not well regulated or documented. The capacity of urban green spaces to support released animals is very difficult to measure, and it is quite possible that the numbers of released individuals could rapidly exceed the carry capacity, especially where multiple operators work in the same area. Little is known about how operators make decisions around the capture and release of nuisance wildlife. Studies relating to human–snake encounters primarily focus on the frequency of encounters and which species are involved, but often do not investigate factors relating to how and where operators capture and release animals [32–35,39–41]. The numbers of problematic wildlife being moved may be high, and not subjected to scientific scrutiny through a lack of follow-up on survival, translocation impacts or habitat suitability [1,41,42]. In Australia, this is despite regulations that aspire to best practice. For example the Queensland legislative criteria state that 'action under the permit [must be] unlikely to detrimentally affect the survival of the animal in the wild' [43]. There are significant knowledge gaps concerning how operators interpret

such vague directives, and training for nuisance wildlife removal is often minimal [31,44]. Certification to deal with venomous snakes in Australia typically entails a one- or two-day course involving minimal instruction on species identification, and some courses focus only on minimal handling training with tongs or a hook [31,45]. Additionally, the certificate of competency relates to the behaviour on the day and often with captive animals that are more docile than wild reptiles [31]. Once licenced, it is up to the individual to decide whether they charge for their services, and at what price [37,38].

In this study, we surveyed operators of nuisance reptile relocation services in Australia. We tested whether the number and frequency of reptile relocation requests varied with the experience of the operator. To determine if the decision-making process regarding releases varied with the experience of the operator, we asked operators (i) what time of day should reptiles be released after capture in principle, and what time do releases typically occur in practice? (ii) to what distances should reptiles be relocated in principle, and what distances are typically used in practice? and (iii) which environmental features are important when selecting a release site? We also asked operators to specify their level of confidence that relocated reptiles would remain at the recipient site chosen. We tested whether the number of injured and rehabilitated wildlife varied with operator experience and type of enterprise. Since operators may encounter non-local reptilian species, we asked how often escaped pets were detected, the number of encounters and how the operator identified a non-local species. Finally, we documented the opinions of operators about the current state of the relocation industry.

## 2. Materials and Methods

A questionnaire was used to understand the scale of the nuisance reptile relocation industry in Australia, and the processes that govern decisions made by operators. Operators were defined as professional snake catchers, wildlife rehabilitators and members of the public who, for example, had moved a snake off the road or their property during the reporting year. Additionally, operators were invited to share their concerns about how the industry is operating, and how it might be improved. The questionnaire was available online via Google Forms from 12 September 2020 to 11 October 2020, with respondents being asked primarily to focus their answers on the period between April 2019 and April 2020. The timeline preceded COVID-19 lockdown measures, to ensure that the dataset represented a 'normal' operating period. Participants were recruited via targeted dissemination on social media, where there is a strongly active community of reptile relocation specialists in Australia. The link was initially posted on a Facebook page associated with the lead author's research project, with encouragement to share the link. A 'snowballing' effect then ensued, with the link being shared on many private pages and various Australian reptile groups via social media, and through word of mouth. The questionnaire was designed to be completed in less than five minutes and comprised 28 questions in four sections (see Supplementary File S1 for the full questionnaire). Pearson's Chi-square tests were used to compare differences between variables (see Supplementary File S2). All statistical analyses were conducted in the R 4.2.1 statistical environment, using RStudio 2022.07.1 Build 554 [46].

### 2.1. Participant Demographics

Our questionnaire included the following multiple-choice questions aimed at gathering demographic information for each participant: 'What is the area of Australia you work in?', 'What type of area do you primarily work in?', 'What is your gender?', 'How long have you been relocating snakes?' and 'What type of enterprise to you operate under?'. We used the length of time in the industry as a representative for experience and enterprise as commercial (charge a fee per service), salary (paid for work), charitable (funded through donation) or pro bono (no charge).

### 2.2. Number, Frequency and Taxa of Reptile Relocation Requests Regarding Experience

To gather baseline data about the nuisance reptile relocation industry, our questionnaire included 'Approximately how many times have you or your employees been called to a property/business/other to attend to a reptile callout over the last year prior to the COVID-19 lockdown (between April 2019 and April 2020)?', 'What percentage of reptile callouts results in a translocation (moving the animal from point of capture)?' and 'In the year prior to the prior to the COVID-19 lockdown (between April 2019 and April 2020), what percentage of your callouts related to the following taxonomic groups (colubrid snakes elapid snakes, pythons, lizards, turtles, crocodiles, other)?. Experience within the industry was also included as a factor.

### 2.3. Decision-Making Process of Reptile Relocation Requests Regarding Holding Time, Time of Day, Distance and Importance of Environmental Features for Release

We included experience as a factor and asked the following questions to understand what is currently occurring within the industry: 'What would be your typical action of a captured reptile?', 'In principle, at what time of the day do you typically aim to release snakes that have been held¿, 'In practice, at what time of the day do you typically aim to release snakes that have been held?', 'In principle, what is the best distance to release snakes from their point of capture?', 'In practice, what is the typical distance snakes are released from point of capture?', 'How often do you use the same release site for relocated snakes?', 'How important are the following criteria when looking for a release site? Proximity to buildings and/or likelihood of human encounter, distance to domestic pets, habitat type, vegetation type and density of cover, bushland or green space, distance to water, potential food availability, and known release site used previously'.

### 2.4. Level of Confidence That Relocated Reptiles Remain at the Recipient Site Regarding Experience and Enterprise

To identify the level of confidence of operators in their choice of release site selection, we asked participants 'How confident are you that the snakes you translocate remain in their new environments?' and used experience and enterprise as factors.

### 2.5. Number of Injured Wildlife and Rehabilitation Regarding Experience and Enterprise

To understand other facets of nuisance reptile relocation, we asked, 'How often of the following happens during a typical reptile call out: animal is caught, animal is not caught, misidentification, wildlife rehabilitation needed, animal is dead or dies shortly after?', 'What typically happens to injured wildlife?' and 'List the three main causes of wildlife rehabilitation if encountered', and we used experience and enterprise as factors.

### 2.6. Number of Encounters with Escaped Pets and Exotic Reptilian Species

As operators may encounter non-local reptilian species, dichotomous questions were used to investigate escaped pet and exotic reptile encounters. For participants that had checked yes to the question, 'Have you ever attended a snake call outs that turned out to be an escaped captive (pet) snake?', we further asked, 'How did you know it was an escaped captive snake?', 'Roughly how many call outs to escaped captive snakes have you had in the last year prior to the COVID-19 lockdown (between April 2019 and April 2020)?' and 'To the best of your knowledge what happened to the snake?'

### 2.7. Opinions about the Relocation Industry

To gain an understanding of operator concerns and viewpoints within the industry, we asked two questions: 'Do you have any concerns regarding snake relocation as it is currently being practiced and/or promoted?' and 'Are there any other comments you would like to share?'. Reflexive thematic analysis was conducted in NVivo 12 Plus using an inductive approach [38]. Initially, each comment was segregated and grouped by similar context, resulting in four themes: animal welfare, perception of the industry by operators, more

general comments and no concerns. Next, each theme was refined into sub-categories, also grouped by context, with categories added as needed. Due to the nature of open questions, most comments covered multiple aspects. If the comment included multiple mentions under the same sub-category, this was counted once. Within each theme, several sub-categories were apparent, and comments were grouped accordingly. The procedure was repeated independently by a second analyst (U.A.M.F.), and any disagreements discussed and resolved. Minor spelling and grammatical errors were corrected for presentation, without changing the meaning of the comment. Responses with 'no' or similar responses to the second question allowing for additional comments were discarded.

## 3. Results

### 3.1. Participant Demographics

A total of 125 people responded to the survey. There were significantly more responses from males than females (93 males, 32 females; $\chi_1^2 = 29.77$, $p < 0.001$) (Figure 1A). Eighty (65%) participants were operating within urban areas, comprising 38 (31%) working in the inner city and 77 (62%) in suburbia. Eighty-four (68%) participants reported operating in semi-rural areas, 52 (42%) in rural, 41 (33%) in farmland and 41 (33%) in bush locations. There were survey respondents from every State or Territory of Australia. Queensland had the highest number of participants (60; 48%), followed by New South Wales (43; 34%).

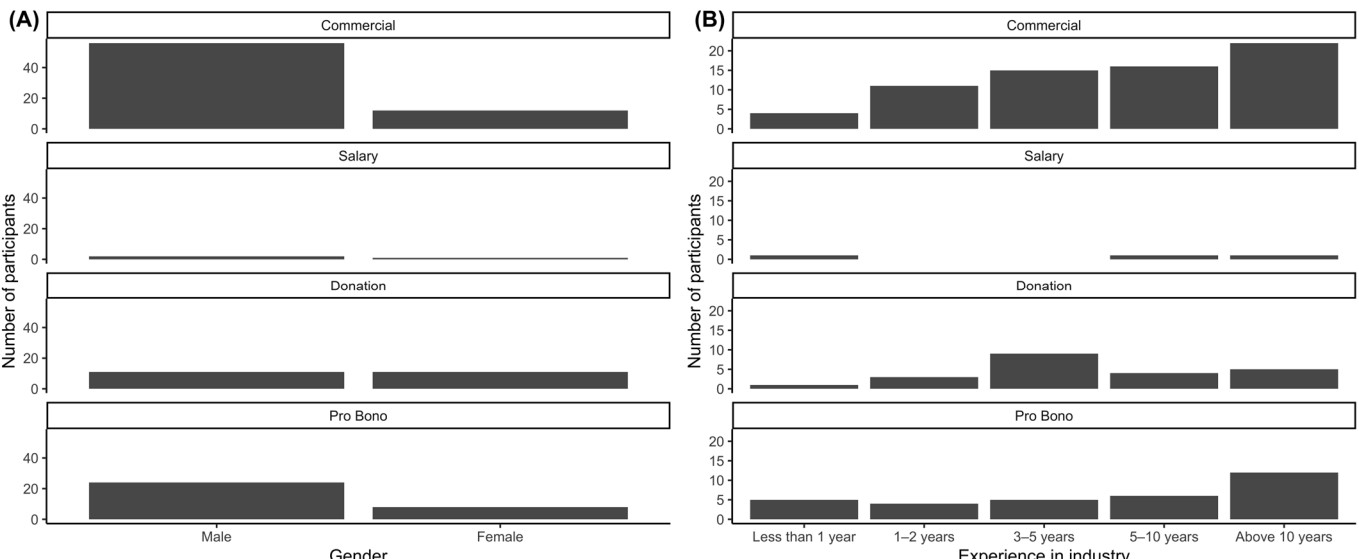

**Figure 1.** (**A**) Study participants broken down by gender and operating enterprise type (commercial (charge a fee per service); salary (paid for work); charitable (funded through donation) or pro bono (no charge)). A gender bias towards males was observed overall but similar gender division for charitable and pro bono enterprises. (**B**) Level of experience for participants based on years in the industry and operating enterprise type. Level of experience was relatively high, and type of enterprise varied between the four enterprise groups, indicating the broad scope of participants.

Seventy-one (57%) participants operated a paid enterprise (68 paid per job; 3 salaried), with remaining participants relocating for donations (22; 18%) or no charge (32; 26%) (Figure 1B). The level of experience in the industry was generally high, with 40 (32%) participants having more than 10 years' experience in relocating reptiles, and only 11 (8%) participants with less than one year (Figure 1B). There was no significant difference in the type of enterprise at different levels of experience ($\chi_{12}^2 = 11.47$, $p = 0.489$).

### 3.2. Number, Frequency and Taxa of Reptile Relocation Requests Regarding Experience

Participants with more years of experience in the industry received a greater number of relocation requests ($\chi_{15}^2 = 29.77$, $p < 0.001$); most participants with less than one year's

experience had fewer than 10 requests over the reporting year (Figure 2A). Most operators received less than 100 requests in the reporting year (this accounted for 73% of participants). Primarily, those operators with more than 100 requests annually had five or more years of experience in the industry, although one operator with one–two years' experience indicated that they received 501–1000 requests. All operators moving >501 animals operated under a commercial enterprise, including two individuals with more than five years' experience who indicated that they had received more than 1000 removal requests during the reporting year (Figure 2A). Although all participants received relocation requests, only around half of operators reported actually relocating animals more than 80% of the time, although this varied somewhat with the level of experience in the industry (Figure 2B). Notably, slightly over a quarter of respondents with less than one year's experience indicated that they relocated animals 0–20% of the time. Taxa eliciting the most relocation requests were elapid snakes (92% of participants had been asked to relocate elapids at some time), pythons (86%) and lizards (79%).

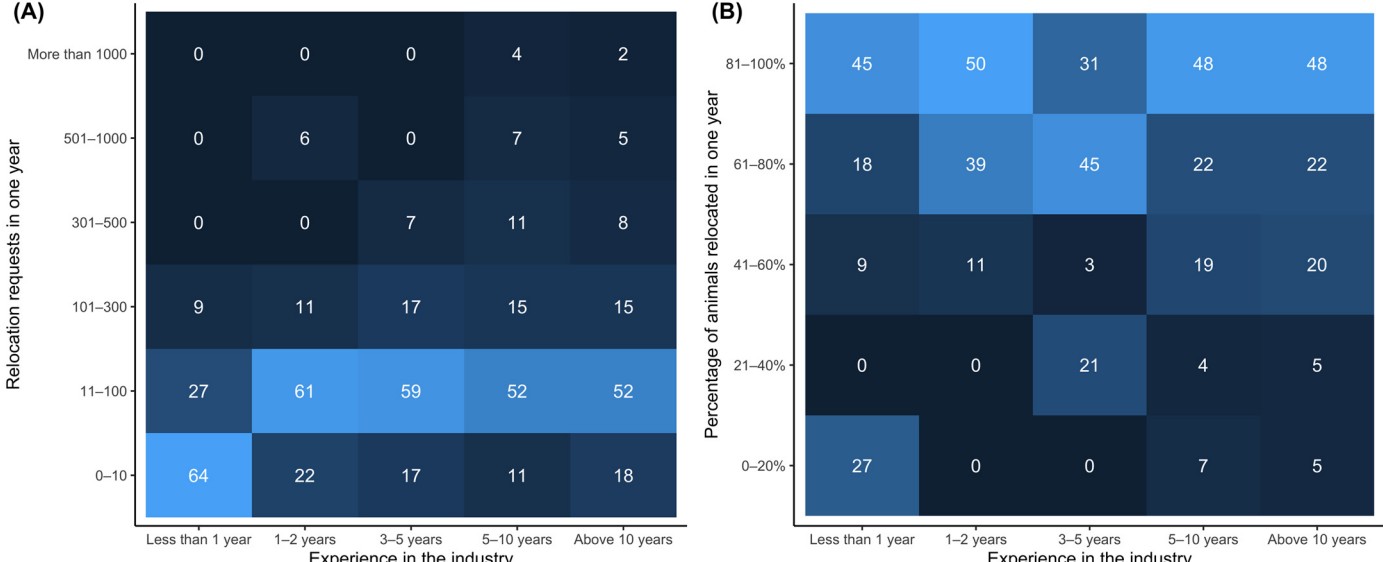

**Figure 2.** (**A**) Reptile relocation requests (*n* = 125) and (**B**) proportion of animals that were relocated by operators (*n* = 125), broken down by years of experience in the industry of the participant. Numbers indicate the percentage of participants with that particular combination of experience and relocation data, with lighter colours indicating a higher number of responses. Most operators had less than 100 requests to relocate reptiles for the year, and those with more than 500 requests typically had more than 5 years' experience in the industry. On average, relocations occurred in 81–100% of requests.

### 3.3. Decision-Making Process of Reptile Relocation Requests Regarding Holding Time, Time of Day, Distance and Importance of Environmental Features for Release

When relocations did occur, 86 (68.8%) participants indicated that they primarily released captured reptiles immediately, while 31 (24.8%) typically held the reptiles until a convenient time for release, and the remaining eight (6.4%) participants held animals overnight to be released the next day. The most frequent response to the question about the time of day at which participants believed reptiles should be released was 'dependent on species' (73 responses, 59%), presumably varying for diurnal, nocturnal or crepuscular species (Figure 3A). Reflecting this belief, 60 participants indicated 'dependent on species' as the time that they physically released captured reptiles, and another 14 (11%) chose this option along with additional release times (Figure 3A). More releases occurred at specific times than the level of belief in those times as being appropriate would suggest. Morning releases were more common than the belief that this time was appropriate, and evening releases occurred less frequently than the belief would suggest (Figure 3A).

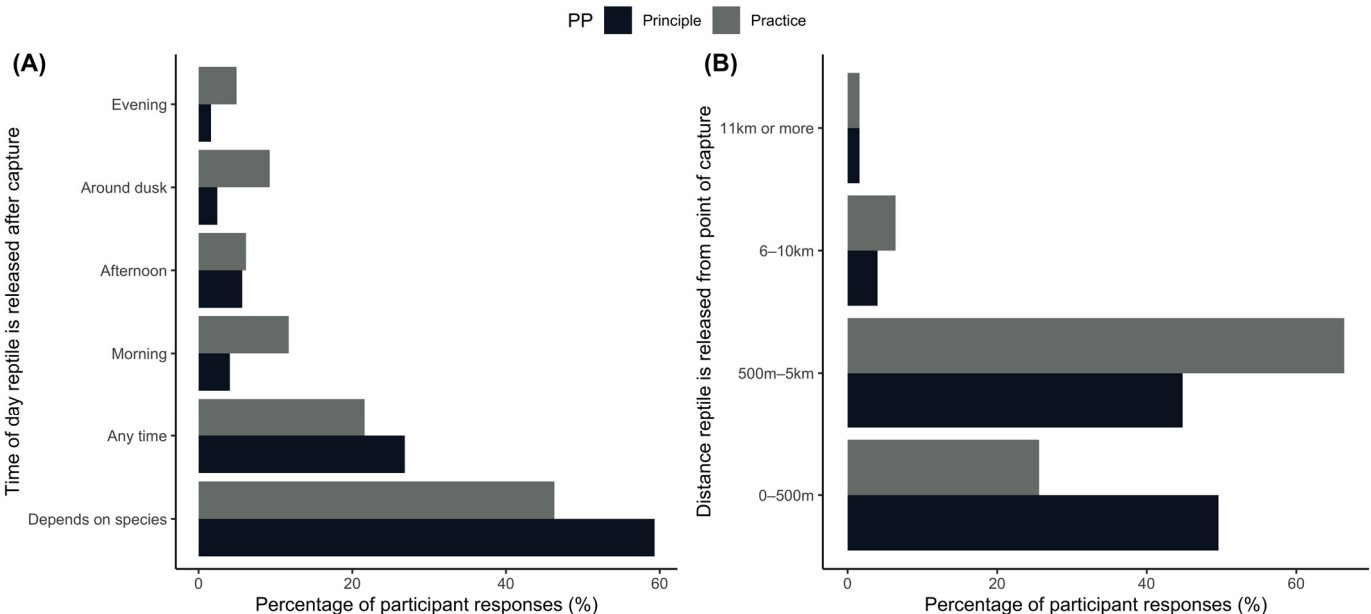

**Figure 3.** Differences in principle (what participants believe should happen) and practice (what actually occurs) during reptile release according to (**A**) time of day (*n* = 123) and (**B**) distance from capture location (*n* = 125).

Distances at which animals were actually released differed even more strongly from participants' beliefs about appropriate distances. The most common belief was that animals should be released within 500 m, but the most common distance at which animals were actually released was 500 m–5 km (Figure 3B—although we note that perhaps this was a survey flaw, perhaps reflecting a lack of suitable options near to the preferred distance category). One participant operating commercially with one–two years' experience believed that long-distance relocations (>11 km) should be used; however, the same respondent specified that they relocated 11–100 reptiles within short distances (500 m–5 km). A second participant, also operating commercially with one–two years' experience, indicated that they believed short-distance relocations should be used (500 m–5 km) but relocated between 101 and 300 reptiles over long distances (>11 km).

Seven of the eight environmental criteria for the characteristics of a suitable release site had the highest number of responses for 'Very important' (Figure 4). Habitat type was the most highly cited criterion, with 73% (*n* = 89) of participants describing it as very important, followed by vegetation type (81; 66%), potential food availability (70; 57%), distance to bushland (62; 53%), distance to domestic pets (61; 50%), distance to water (59; 49%) and distance to buildings (56; 46%). The use of known release sites had the highest response for 'no preference' (38; 31%) but the second-highest response was 'important' (32; 26%; Figure 4). These results indicate that all the suggested criteria are considered in some form before reptiles are released at a recipient site.

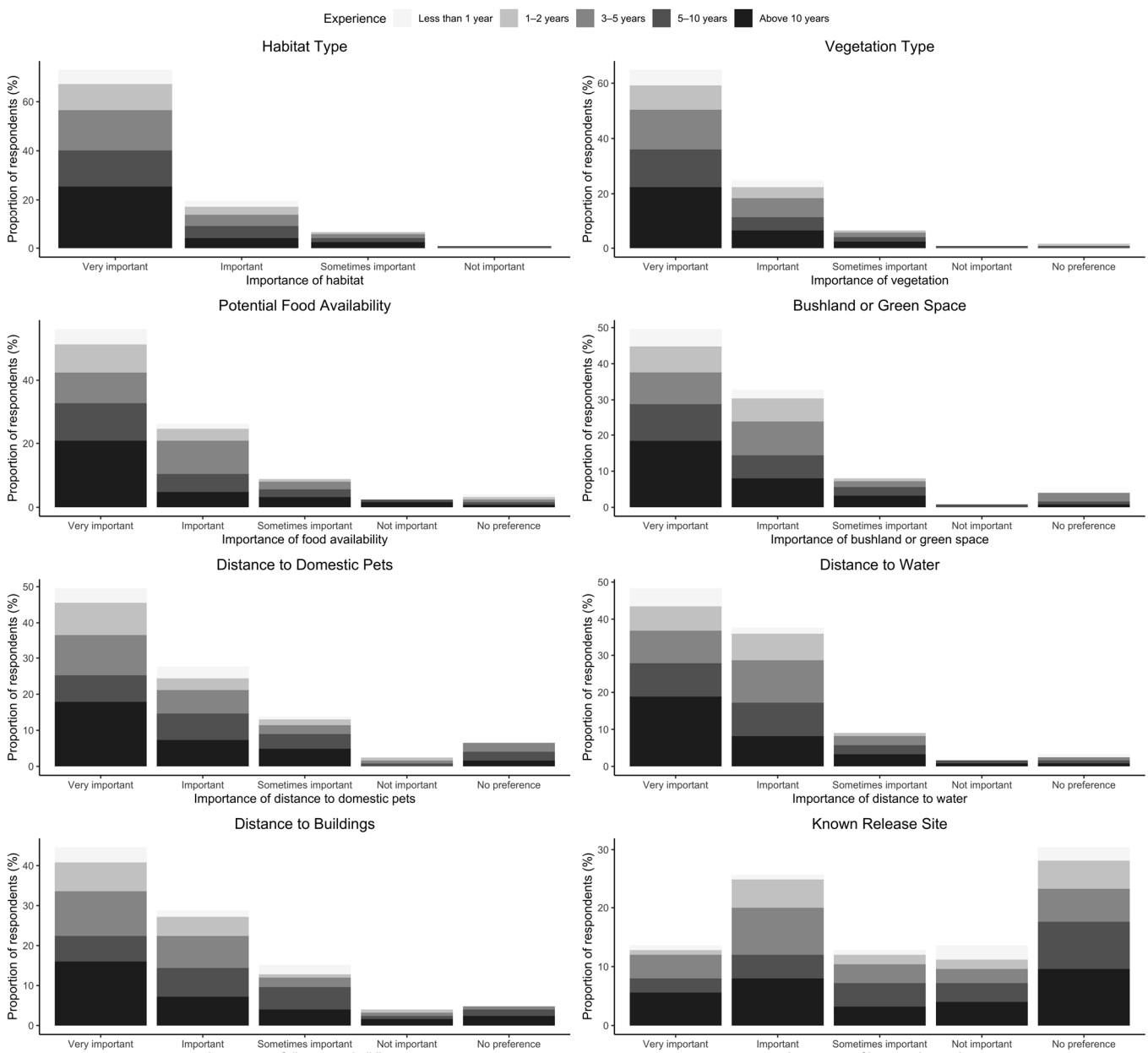

**Figure 4.** Importance of eight environmental features when selecting a release site for relocated reptiles by participants with varying degrees of experience. All environmental features, apart from 'known release site' were scored as very important by more participants than any other importance level.

### 3.4. Level of Confidence That Relocated Reptiles Remain at the Recipient Site Regarding Experience and Enterprise

Fifty-seven (45.6%) participants indicated they were 'undecided' on whether relocated reptiles remained at the recipient site. However, 41 participants (32.8%) were confident and 14 (11.2%) were 'very confident' that the animals would remain at the recipient site. In contrast, 11 (8.8%) participants were 'not confident' and two (1.6%) participants were 'not confident at all', suggesting variability in the degree of certainty among operators about the potential outcome of their relocation efforts. Participants who indicated 'not confident at all' had three–five years' experience and operated under commercial and charitable enterprises, relocating 0–10 and 11–100 animals per year, respectively. Of the eight participants who relocated more than 500 reptiles per year, five indicated that they were 'undecided' on whether the animals remained at the chosen recipient site. Analysis showed no significant

statistical interaction between confidence rates and years of experience ($\chi_{16}^2 = 18.556$, $p = 0.292$).

### 3.5. Number of Injured Wildlife and Rehabilitation Regarding Experience and Enterprise

Encounters with injured wildlife (due to causes detailed below) were reported by all types of enterprise (*n* = 123), with wildlife rehabilitation being needed 'not often' (84; 68%), on 'average' (22; 18%), 'quite often' (10; 8%) and 'never' (7; 6%). Apart from salaried positions, and those with less than one year's experience in humanitarian and charitable enterprises, wildlife rehabilitation was provided by 27 (22%) participants for all remaining levels of experience and enterprise types. The main causes of injury mentioned were domestic pet encounters (83; 33%), human interference (66; 27%), vehicle or road injury (52; 21%) and entanglement (24; 10%). A poor body condition was reported by 23 (9%) participants. Only 19 (15%) of 122 responses chose 'never' regarding how often an animal dies after capture, although we are unable to determine whether deaths typically occurred because of injury prior to capture or resulted from the capture itself.

### 3.6. Number of Encounters with Escaped Pets and Exotic Reptilian Species

Relocation requests involving escaped native captive (pet) snakes were reported by 81 (65%) participants, totalling more than 370 fugitive individuals. Participants indicated that they diagnosed the animals as escapees because they were not native to the region in 72 (89%) cases, were of a domestic phenotype in six (7%) cases or were advertised as missing by owners in three (4%) cases. Although illegal to keep in Australia, exotic non-Australian reptile species were encountered by 32 (26%) of participants, with 20 (63%) located in New South Wales, nine (28%) in Queensland, two (6%) in Victoria (6%) and one (3%) in the Australian Capital Territory. Approximately 118 individual exotic snakes were encountered during 2019–2020, with more than 415 during participants' time in the industry. All exotic reptiles were reported to the wildlife authorities for each state. One participant in inner-city New South Wales indicated that they had been called out to 'hundreds' of exotics during their time in the industry.

### 3.7. Opinions about the Relocation Industry

A total of 137 responses were received from 89 (71%) participants for the two optional open-ended questions. The opening question about whether the participants had concerns about the relocation industry received 84 (61.3%) comments and the opportunity for further comment attracted 41 (29.9%) responses. Nine (6.7%) comments stated 'No', 'Not in particular' or 'No the training is thorough', and three (2.1%) indicated 'Yes' but did not elaborate further regarding concerns about the industry. The remaining 125 comments were distilled into 185 mentions, covering two themes and 10 categories (Figure 5; see Supplementary File S3 for all comments). Comments in the eight largest categories are summarised below.

#### 3.7.1. Ecological Knowledge (Welfare Theme; 37 Mentions)

Ecological knowledge was perceived as highly variable among operators; for example, it 'is very reliant on the actions and experience of the individual which in my opinion can be highly variable'. Perception of what is best for the snake being relocated ranged from an ecological point of view, e.g., 'Removing snakes from the location they were found affects the local population numbers and overall balance. Snakes moved from their local area lose shelter sites and very often die. I have a policy of releasing the snakes as close as possible to where they were caught, often right next door to where they were found. A snake found in an urban area was obviously living and thriving happily in that environment—it naturally occurred there and therefore the presumption that removing it from that urban environment on release is wrong', to conflict with what the general public are seeking; for example, 'I know what is best for the snake but this is often times at odds with what is practical or what the public expects to be done'. A broader welfare concern regarding the

impacts of the relocation industry was expressed as 'I believe a great number of snakes are relocated and then ultimately rescued again after suffering the effects of being unable to find food or water'.

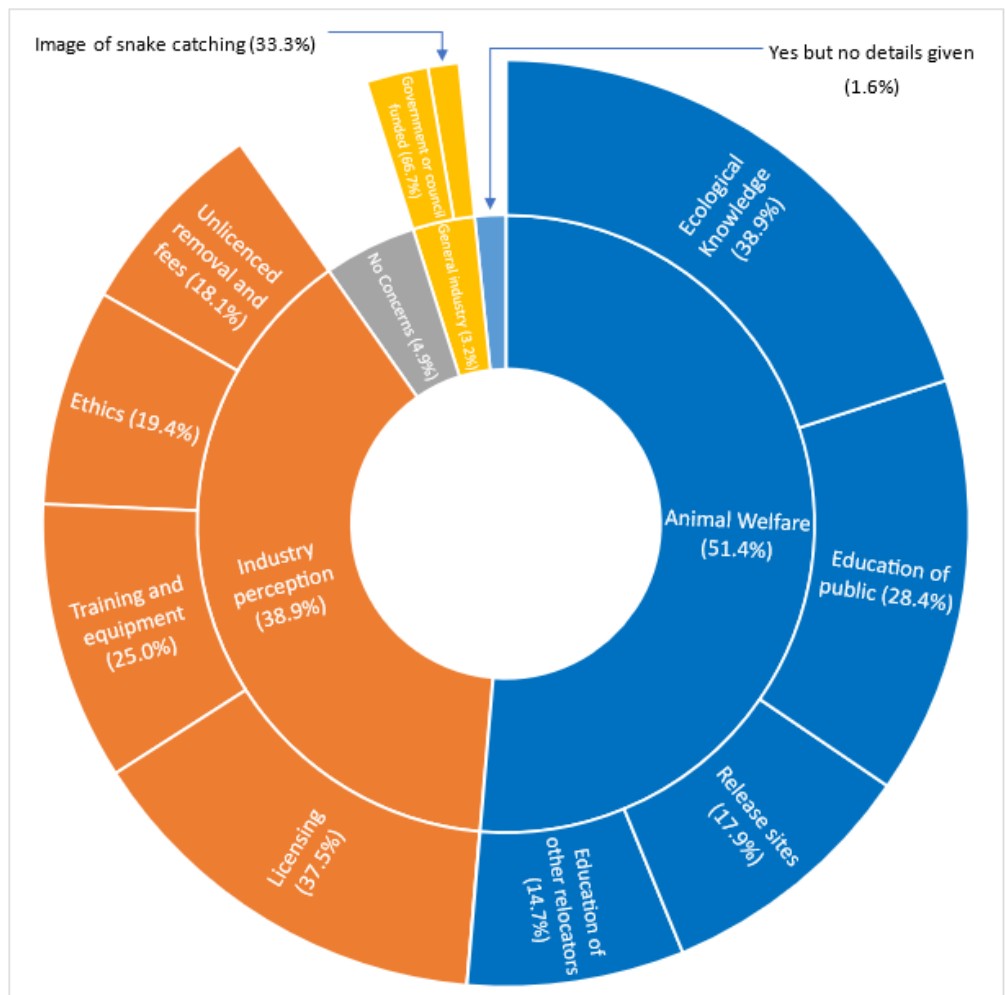

**Figure 5.** Eighty-nine operators provided responses to the questions 'Do you have any concerns with the way the industry is portrayed?' and 'Any other comments?'. Each comment was separated into themes (inside ring) and categories (outside ring). The parentheses indicate the percentage of comments relating to each theme and category.

### 3.7.2. Education of Public (Welfare Theme; 27 Mentions)

Education of the public was the second-highest category with the desire for improvement in terms of not only reducing the need for relocations to happen but also for the welfare of the reptiles being relocated. For example, 'Improving public education is key to reducing number of snake translocations which I suspect is having adverse implications' and 'There needs to be more public education about snakes. Many call outs are only to find the snake stressed as it has been harassed and, in some cases, injured from people trying to kill them'.

### 3.7.3. Release Sites (Welfare Theme; 17 Mentions)

Uncertainty among operators regarding distance and ethics to relocate snakes was apparent; for example, 'I understand that they should be released as close as possible to the capture site, but think that it is in the reptile's best interests that they be released in a remote location (within reason) to lessen the likelihood of human encounter'. Additionally, this occurred even within individuals, e.g., 'two fields of view. Release all snakes including highly venomous very close to capture point or release all snakes caught well away from

human habituation. I prefer to release highly venomous snakes further away and all other reptiles very close to point of capture'. Concerns regarding moving urban snakes to bushland were raised; for example, 'My concerns are with the lack of understanding of the ecological requirements of these snakes by many snake catchers. For example, taking urban snakes which are living in areas with high densities of food and taking them to bushland reserves with inappropriate habitat and lower densities of prey items. These animals need to stay in the areas in which they were captured and released at sites that meet individual species requirements'. Differing acceptable relocation distances between legislative areas of Australia were questioned; for example, 'ACT requires snakes to be released within 5 km and NSW 20 km, one has to be right meaning the other is wrong'. Those from Western Australia specifically argued for a reversal in the recent abolition of reptile relocation licences, e.g., 'Here in WA, they have recently changed the rules so that you do not require a licence to catch and relocate a snake. I think that's asking for trouble for both snake and human. Would like to see this reversed'.

3.7.4. Education of Other Relocators (Welfare Theme; 14 Mentions)

Animal welfare concerns regarding the perception of how other relocators operate highlight a lack of regulations within the industry, e.g., 'Some catchers clearly want to be engaged for every scenario (outside snakes, pythons that are near boundaries) & don't encourage residents to understand their habits & traits' and 'My concerns are with the lack of understanding of the ecological requirements of these snakes by many snake catchers. For example, taking urban snakes which are living in areas with high densities of food and taking them to bushland reserves with inappropriate habitat and lower densities of prey items. These animals need to stay in the areas in which they were captured and released at sites that meet individual species requirements'. Concerns also related to the lack of incorporation of injured wildlife into industry obligations, e.g., 'The catchers that pounce on a paid job but refuse to go around the corner for injured/rehab wildlife'. Additionally, conflict between charity and paid relocators was apparent, 'There are many unlicensed snake relocators and relocators with little or no training, such as emergency services (fire and rescue, VRA, SES, police) and wildlife rescue group members'. Furthermore, concerns regarding animal welfare arose from this conflict, 'organisations such as the fire brigade doing it, when they have no knowledge about the animals, and different species, and are rough with animals, using tools to capture which may hurt animals, as they are usually scared of them, and don't really care for the animals, as it's just a part of their job these days'.

3.7.5. Licensing (Industry Perception Theme; 27 Mentions)

Concerns related to the ease of obtaining permits ranged from 'instantly computer automated with no human processing' to 'the criteria requests a first aid certification be held by an applicant. This appears to be an attempt to address the safety concerns that coincide with the regular handling of elapid snakes. At no stage are detailed risk assessments or handling competencies reviewed with regards to venomous snake interaction. The unmitigated movement of our wildlife via this permit as opposed to the myriad of hoops that need to be jumped through via ethics and scientific approvals for well credentialed and highly competent researchers and/or professionals is stark. This permit is issued without any definitive question as to the welfare of the fauna involved and undermines the values presented by other permits/approvals for fauna interactive activities issued by the same department.' Specific suggestions regarding changes to the industry were given; for example, 'Yes, I think there needs to be a nation-wide protocol. Including the professionalism of the relocators themselves e.g., safety gear, uniform, price, insurance etc. Snake relocators need to be viewed as a more professional practiced and respected as such. Cleaning up the current industry would do that or would put us all on the right track'. Additionally, the concept of government or council funding was raised by three operators; for example, 'I feel more snakes would be saved if snake removal was government funded or council funded'.

### 3.7.6. Training and Equipment (Industry Perception Theme; 18 Mentions)

Many operators believe that others within the industry have inadequate identification skills and a lack of basic biological knowledge about the animals due to the lack of training standards, e.g., 'my main concerns are the governing body giving licences to anyone that can tick a box, several "catchers" do not know species, behaviours and results in captive and non-native animals being released and the wrong information getting passed on' and 'The holding of a permit to conduct these activities should be restricted to individuals who can clearly demonstrate sound ecological knowledge and long-term safe handling of snakes of all species. The two-day course offered by some companies should be viewed as contributory to an application but to suggest competency is achieved in such time is akin to handing someone an electrical ticket for displaying competency in handling the change of a light bulb'. All concerns with equipment were related to the use of tongs, e.g., 'I have noticed a rise in catchers using tongs and not using proper techniques for handling snakes'.

### 3.7.7. Ethics (Industry Perception Theme; 14 Mentions)

A need for a better scientific underpinning for industry practices was indicated, e.g., 'I believe we are causing more harm than good. The unfortunate part of this industry is there is not enough research and information out there to make a proper informed decision into the welfare of each individual species'. Concerns with how the industry currently operates without comprehensive regulations were also raised: 'it's become a lucrative and unpoliced industry leading to unethical and illegal practices'.

### 3.7.8. Unlicenced Removal and Fees (Industry Perception Theme; 13 Mentions)

Conflict is occurring within the industry regarding no regulations or standardisations for fees, e.g., 'Just that it's very difficult these days to be a full-time professional catcher, and be respected, and have the public understand it being a job to get paid for doing, if providing a professional service, and be available nearly always all the time. As wildlife care groups have given the image to the public that they are available free of charge at all times to catch healthy reptiles from scared people's homes, instead of the truth, which they're actually only meant to care for injured wildlife as a volunteer. And being abused by the public for charging a fee is something we as professional catchers deal with all the time these days because of this'. Additionally, operators have concerns with the ability to set their own price for the service, e.g., 'people charging exorbitant fees, then customers killing snakes cos its cheaper'.

## 4. Discussion

Our study has spanned the biological, socio-economic and political facets of environmental social science through the comments and opinions of operations within the reptile relocating industry. More than half of operators reported relocating animals more than 80% of the time, and the decision to relocate was not strongly correlated with the amount of experience that an operator had in the industry. Previously, commercial snake relocators indicated that up to 50% of requests for a relocation were resolved over the phone—for example, through providing information or identifying the animal as a lizard and deeming relocation not necessary [37].

Multiple commercial operators within this study indicated that they relocated more than 500 animals a year. Operator comments suggested that this emphasis on physically relocating animals arises from the strongly negative public perception of reptiles in the vicinity of human dwellings, and their need to feel safe. Indeed, more than 85% of 11,067 snakes 'rescued' by a wildlife rescue organisation in Australia over a ten-year period occurred because a snake was identified as being in an 'unsuitable habitat', frequently a euphemism for near or in a human dwelling [41]. Snakes of Namibia relocated 509 snakes over three years, with only 46 (9%) from inside buildings, the remainder primarily from gardens and roads [32]. Moreover, of the 5210 snake call-outs in Darwin over seven years, 94% were for species harmless to humans [35], while, of the snakes 'rescued' by an organisation in New

South Wales, 25% were harmless and 68% were medically significant, although the authors noted a bias towards large snake species in 'unsuitable habitats' [41].

Relocations of non-reptilian 'nuisance' species are tightly regulated in most jurisdictions around Australia. For example, possum relocation is mandated to occur between 25 m and 150 m, depending on which region of Australia the licensing occurs in, with many regions of Australia having additional requirements, such as release at dusk and in an area with something close by to climb, for acceptable animal welfare outcomes (Supplementary File S4). Only three (of eight) jurisdictions in Australia have a restriction on how far reptiles can be released, with maximum distances varying between 2 km and 20 km. New South Wales regulations request that reptiles are released 'out of sight of the public', and South Australian rules stipulate that releases occur away from occupied buildings [47,48]. Additionally, in South Australia, Tasmania and Western Australia, it is legal for members of the public to kill snakes if they feel threatened (Supplementary File S3). The lack of consistent regulations for reptile welfare was a concern raised by several operators, with comments relating to a lack of training, negative impacts, moving urban reptiles to bushland and the prioritisation of money over animal welfare. Some operators reflected that this lack of priority for reptile animal welfare indicates a need for operators and the public alike to reconsider how best to deal with the human response to reptiles in close proximity to human dwellings, and requested more scientific studies to understand the ecology and impact of the animals being relocated. One report investigating snake removal trends indicated a gradual decline in encountering venomous snakes in suburbia [35]. Site clearing for archaeological preservation reported declines in two previously common species over 10 years [49]. With other studies showing increasing trends per year for snake encounters, such declines may indicate the beginning of localised extinctions, and highlight the need for data recording and analysis [32,35,41,50].

Over two thirds of operators (69%) indicated that they released captured reptiles immediately, while a quarter held animals until a convenient time for release. Under state restrictions, Queensland requires same-day release, South Australia within 24 h and New South Wales within three days (Supplementary File S4). Although no Fauna Taking (relocation) licences are required for reptile relocations in Western Australia (Regulation 50), releases must occur within six hours; otherwise, fines may occur [51]. Many operators reported releasing animals at times of the day that they regarded as suboptimal; for example, morning releases occurred more commonly than believed appropriate, and evening releases less frequently. Most operators indicated that they released animals immediately, although the high number of morning releases potentially suggests that animals caught in the evenings are being held until the following morning. Indeed, one comment suggested that operators are holding on to snakes for longer periods of time to release at their convenience or in batches.

Release distances were higher in practice than many operators believed to be appropriate, although there was some disagreement about the optimal distances for release. Some operators perceived that snakes should be relocated well away from people, either to reduce the risk to people or to reduce the chance of the animal being relocated again. Other operators believed that the snake should be released close to where it was captured. The average block size in suburban Australia is 467 m$^2$ [52], which is 0.047 ha. Home range sizes and occupancy patterns are not well understood for most reptile species, and might vary markedly over space and time. Commonly encountered venomous species in urban areas, such as tiger snakes *Notechis scutatus* in a suburban parkland, maintained home ranges of 3.88 ha [53], whereas male eastern diamondback rattlesnakes *Crotalus adamanteus* on a university campus sustained home ranges of 65.7 ha [54]. In outer suburbs, male diamond pythons *Morelia spilota spilota* also exhibited larger home ranges of 52.8 ha, 47% larger than the female diamond python's home ranges [55]; however, in a rural environment, home ranges were slightly smaller at 41 ha [56]. Eastern brown snakes in a semi-rural area had home ranges of 5.8 ha [17]. Coastal carpet pythons *Morelia spilota mcdowelli* in semi-rural Australia had a 71% smaller home range than rural diamond pythons [57]. Suburban

eastern blue-tongue lizards *Tiliqua scincoides* maintained different-sized home ranges based on sex, with females exhibiting a home range of 0.51 ha and males 1.27 ha [58]. Urban male Gila monster *Heloderma suspectum cinctum*'s mean home range is 66.2 ha, which is similar to that of rural male lace monitors *Varanus varius* of 65 ha [59–61]. At any point when encountering an animal, it is uncertain where in the home range the animal is located; thus, even relocations of 1 km may take them outside their home range or familiar area.

Removing reptiles from an area does not necessarily prevent the animal from returning [53,62–67]. Short-distance translocations (SDT), defined as within an animal's home range [68], are now being investigated to mitigate temporal contact with humans [69]. Suggested values range from 0.5 km [69] to 1.5 km [31]. Sixteen percent of 39 reptile translocations occurred due to human–wildlife conflict, instead of conservation or research reasons, and these had high failure rates [34]. For example, twelve western rattlesnakes *Crotalus oreganus* moved 500 m to undisturbed habitats returned to the original urban landscape within 28 days, which was considered a failure to mitigate the conflict with humans [69]. A Malayan krait *Bungarus candidus* caught adjacent to an urban dwelling was relocated on two occasions to a forest around 150 m away but returned to the same urban dwelling within a day [39]. Although strong quantitative studies are lacking, at least some anecdotal reports suggest that high mortality after commercial translocation may be commonplace. Three dugites *Pseudonaja affinis* relocated 200 m from an urban area to bushland were all killed within 12 days by a car strike or predation, and an additional four dugites translocated 3 km perished within 6–49 days, also due to predation or car strikes [70]. Only 38 (8%) of 464 marked snakes were encountered again after relocation in the city of Darwin, and six were found dead or required euthanasia due to human involvement [31]. Long-distance translocations (LDT) for snakes are those occurring outside the snake's activity range or twice the distance of travel within a year, moving the animal outside its home range or familiar area [68,71]. Snake long-distance translocations have been variously classified as >3 km [70,72], 9 km [73], up to 30 km [74], 40 km [75], 60 km [76] and 172 km [77]. Four snake rescuers estimated the rescue of 40,500 nuisance snakes in India over 13 years, with relocation distances occurring between 15 km and 60 km [78].

In this study, we found that over half of the operators considered previously used release sites as a guiding factor in choosing future release sites, suggesting that many reptiles are being released at the same site by any given operator. Multiple releases of species at the same site suggest potential for the saturation of animals, leading to competition for resources and territory [8]. Operators highlighted a concern that individuals within the industry are using the same locations for multiple releases, regardless of the species' habitat requirements, and leading to concerns about multiple operators practicing in overlapping areas. Previously, two Australian operators indicated that they released approximately 20 and 250 animals per year at separate, previously used locations [38]. With limited industry regulations, there is an argument that the industry is becoming flooded with multiple catchers operating in the same area, with little thought about how this could impact carrying capacity or animal welfare.

Confidence among operators about reptiles remaining in the chosen recipient site was low, and several operators indicated a desire to better understand the fate of relocated animals through scientific research. The habitat type for recipient sites was generally considered very important by operators, although there were opposing views about which sites should be used for releases; some operators had a belief that snakes should be moved to bushland away from human habitation, while others suggested that snakes should be released near to where they were found. The variation in release site choice highlights the need for scientific studies to inform regulations. 'Suitable habitat' was listed as the release criterion for studies involving tiger snakes and timber rattlesnakes *Crotalus horridus*, along with the additional requirement of an active hibernaculum used by resident rattlesnakes [77]. Venomous nuisance snakes in Darwin were relocated to either the closest patch of natural bushland [35] or outside the town's boundaries [31]. Non-venomous snakes were released in the closest greenspace as bushland or parkland, typically <500 m,

or the closest uninhabited land or conservation reserve between 1 km and 2 km with immediate sheltering sites [31,35]. Resident urban dugites were released in bushland <200 m from the capture site, whereas relocated dugites were set free in bushland, meeting the criteria of native ground and canopy cover with visible coarse wood debris [70]. From four studies looking at the relocation of rescued nuisance king cobras *Ophiophagus hannah*, two studies had no mention of the habitat criteria for release [76,79], one released them as close as possible to the capture site [33] and one study released them within a fragmented rainforest with human habituation [78]. Relocated nuisance Gila monsters were released adjacent to packrat nests or rodent burrows as appropriate refuges [67]. White-lipped pit vipers *Trimeresurus albolabris* were released in mixed shrubland/grassland/abandoned agricultural pond habitats [72]. Western diamond-backed rattlesnakes *C. atrox* involved in SDT were released at sites based on stratified random sampling, which avoided heavily used human areas [71]. Field studies show that released animals often shift their habitat use, perhaps because of competition from residents. Translocated red diamondback rattlesnakes *Crotalus ruber* increased their use of vegetation and decreased their use of rocky habitats compared to resident snakes [69], and translocated northern water snakes *Nerodia sipedon sipedon* spent more time in open wetlands than residents who spent equal amounts of time in open and forested wetlands [66].

Only a few operators indicated that animals required rehabilitation on a regular basis, yet, over an 11-year period, one wildlife hospital received 31,626 admissions, with reptiles comprising 14.4% (4568) and snakes 4.5% (1426) [50]. Often, the lead author is contacted by commercial operators seeking advice for injured reptiles. Since injury, identification and management is not always comprehensively covered in training, it is possible that many animals with injuries are being released unintentionally. It is our belief that reviews and updates of training practices are needed to deal with the crossover of humans and animals existing in the same environment and injury should not be reliant on wildlife rehabilitators alone.

Current training practices overlook the identification of non-endemic native species popular in the pet industry, and exotic species. The lack of identification capabilities within the industry was a concern amongst many operators as an increasing threat to native species. All operators within the industry should be familiar with their local species and able to identify escaped pets and exotic wildlife, to be handed in to relevant authorities. Threats from non-endemic and exotic reptile species include novel diseases [80,81], parasites [82–85], competition for resources causing displacement [83] and hybridisation [86]. More than half of our study participants indicated involvement with a native species not occurring in the local area, presumably escaped pets. Most of the escaped pets were either given back to owners, handed to authorities or rehomed. Escaped, unwanted and seized pet reptiles comprised 83% of non-local encountered native species in NSW [87]. Of the 143 non-local carpet pythons encountered in 2017, 17% (25) were hybrids from captivity and five were unable to be assigned to a clear species heritage, highlighting the potential relocation and release of captive animals [87]. More than a quarter of participants encountered exotic species, with more than 100 individual encounters for one year. Known exotic wildlife smuggled into Australia, as part of the illegal pet trade, include red-eared sliders *Trachemys scripta elegans*, ranked among the worst 100 invasive species, and boa constrictors *Boa constrictor* and corn snakes *Pantherophis guttatus* [82,88]. In the Sydney area, red-eared sliders have confirmed alien-established populations and corn snakes have been detected through accidental or deliberate release from captivity [82,89]. Red-eared sliders have also been detected in Queensland, as established populations or being kept illegally, fuelling the possibility of deliberate release (Mo, 2019, DAF unpubl. data). Furthermore, Biosecurity in Queensland intercepted an additional eight exotic reptile species between 2017 and 2021, totalling 30 individuals, either at large or through illegal keeping (DAF, unpubl. data). Comments during our survey highlighted concerns regarding a lack of identification skills within the industry—clearly, it would be preferable from a biosecurity perspective to remove escaped pets and non-native species animals from the wild.

The opinions and comments gathered within this study suggest a real need to understand how the reptile relocation industry is operating with a view to incorporating stronger government regulatory measures and highlight concerns amongst operators themselves, especially regarding animal welfare. Many operators called for increased training and professionalism within the Australian reptile relocation industry, echoing a similar pattern in the US, where more than three quarters of nuisance wildlife operators supported certification and licencing with the additional requirement of public liability insurance [9]. Operators within the industry are calling for a collaborative endeavour to move away from the traditional 'snake catcher' model, and towards 'snake consultants' [45]. Under this model, snake consultants would provide the opportunity for informed choices by the public, where the objective is to create an outcome that benefits the client and the animals' welfare and an understanding of how their requests will impact the fate of the wildlife, rather than relocating an animal for payment [45,90]. Additional obligations for the industry should also include the following: developing knowledge about the focal species and the surrounding environment, general disease considerations of relocations and release-specific considerations [91]. Careful planning and attention to knowledge gaps are needed, especially due to the lack of basic information on the natural life history of many species, including the temporal and spatial use of the landscape [67,92]. Under the IUCN translocation guidelines, relocations should address animal welfare implications including the following: deciding when a relocation is an acceptable option, biological (animal) and social (human) feasibility, conducting a risk assessment, monitoring and continuing management and the dissemination of information [93]. We believe that embedding the IUCN guidelines into industry regulations would be a useful first step towards addressing animal welfare issues, including the removal of female snakes from nests during breeding season and from ceiling cavities during winter, which is currently lacking throughout all jurisdictions. Species-specific or taxonomic group regulations based on scientific literature need to be considered for government guidelines.

A related suggestion from operators was to employ a system such as that used in Belo Horizonte, Brazil or Darwin, Australia, where employed mediators are used for human–reptile conflict [31,34,35]. Those in socio-economically disadvantaged neighbourhoods are more likely to try to deal with the issue themselves, usually by killing the offending animal, while those in more affluent areas will call a professional to remove the offending animal [32,34,94,95]. An employed mediator, through government or local councils, removes the financial component and may improve accountability, monitoring and transparency within the industry [28,96,97].

## 5. Conclusions

The impacts of operators on the animals that they relocate remain essentially unstudied, and arguably the industry has grown large without a strong scientific basis on which to interpret its ethics. This review has demonstrated high levels of uncertainty from professionals about the functioning of the industry, including a number of significant human and animal welfare concerns. We conclude that re-branding snake catchers as snake consultants, in conjunction with government or council endorsement, might encourage informed choices among the public around the management of nuisance reptile species. Supporting and auditing the process through good governance, as well as monitoring the ecological outcomes through scientific research, is also essential. There seems to be a strong case for policies that standardise training among jurisdictions, conduct research on the impacts of relocations and their interactions and effectively monitor post-release survivorship. Reform of the industry would seem to be in everybody's interest—and perhaps, most importantly, ensure the welfare of the relocated animals themselves.

**Supplementary Materials:** The following supporting information can be downloaded at: https://www.mdpi.com/article/10.3390/d15030343/s1, Supplementary File S1: Reptile relocation questionnaire; Supplementary File S2: Variables used in this study for analysis; Supplementary File S3:

Reptile relocation comments from participant; Supplementary File S4: Australian state regulations for mammals and reptiles. References [98–107] are cited here.

**Author Contributions:** Conceptualization, C.M.D.; methodology, C.M.D.; software, C.M.D.; validation, C.M.D. and R.A.F.; formal analysis, C.M.D.; investigation, C.M.D.; resources, C.M.D.; data curation, C.M.D.; writing—original draft preparation, C.M.D.; writing—review and editing, C.M.D. and R.A.F.; visualization, C.M.D.; supervision, R.A.F.; project administration, C.M.D.; funding acquisition, C.M.D. All authors have read and agreed to the published version of the manuscript.

**Funding:** This research was supported by an Australian Government Research Training Program Domestic Stipend Scholarship to C.M.D.

**Institutional Review Board Statement:** The study was conducted in accordance with Australian Human Ethics Research and approved by the Human Research Committee of the University of Queensland (protocol code SBS/032/19 2020001933 and date of approval 2019).

**Informed Consent Statement:** Informed consent was obtained from all subjects involved in the study.

**Data Availability Statement:** The data presented in this study are available on request from the corresponding author. The data are not publicly available due to human ethics requirements.

**Acknowledgments:** We would like to acknowledge Gillian Paxton for the assistance in designing the questionnaire and John Hall for edits throughout the study. We also thank three anonymous reviewers for their comments on the manuscript and all participants who completed the questionnaire.

**Conflicts of Interest:** The authors declare no conflict of interest.

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
