# Peer review of "The Reptile Relocation Industry in Australia: Perspectives from Operators"

_diversity, doi:10.3390/d15030343_

Round 1
Reviewer 1 Report (Previous Reviewer 2)
Dear authors,
I see an improvement in your manuscript. However, there are formatting issues, particularly the formatting of the figures.
Thanks,
Author Response
Dear authors,
I see an improvement in your manuscript. However, there are formatting issues, particularly the formatting of the figures.
Thanks,
Response 1: We thank the reviewer for their comments and have provided higher resolution figures.
Reviewer 2 Report (Previous Reviewer 3)
Dear
I believe that the authors have carried out a profound remodeling of the article and have modified those suggested items or have suggested a new orientation.
I believe that finally it is a Diversity magazine article with a high level of quality. In its current form it is easy to read and understand. The application of certain suggested methodologies and the performance of new analyses and graphic support make this study understandable and clear in its current form.
My opinion the work can be published without modification.
All the best
Albert Montori
Author Response
I believe that the authors have carried out a profound remodeling of the article and have modified those suggested items or have suggested a new orientation.
I believe that finally it is a Diversity magazine article with a high level of quality. In its current form it is easy to read and understand. The application of certain suggested methodologies and the performance of new analyses and graphic support make this study understandable and clear in its current form.
My opinion the work can be published without modification.
All the best
Albert Montori
Response 1: We thank the reviewer for their comments.
Reviewer 3 Report (Previous Reviewer 1)
Dear Editor and authors,
The text has improved much, yet important concerns remain concerning its structure and graph formatting (the resolution is too low, and the text looks unformatted). As assistant editor of the journal, I praise the manuscripts for reaching an excellent writing level. Despite the additional work for everyone, it makes the article shine and makes potential future authors more willing to contribute to the journal and cite it, which benefits everyone.
The following hints should cover the problem.
Line 109.
When you say we surveyed..or..we investigated, you enter in “methods” language.
Keep on “Introduction” language and transform your objectives into questions or hypotheses you actually answered and/or tested in your methods. This is just fundamental for the reader to know what you did and assess if you did it right.
Example
(i) we estimate the number and frequency of reptile relocation requests
Transform to:
“We asked/tested whether the number and frequency of reptile relocation requests would vary with the experience and gender of the operator and across different population segments.”
See? In this way, you included the factors described in your methods and presented in your results. If you entered a test, state the hypothesis as part of your objectives. It is essential that the crucial variables tackled in each question have been introduced, and their obtention been described, reported and discussed. Otherwise, a lot of unuseful information blurs the important pieces composing the final message.
Then, please remember these questions/objectives/hypotheses in your methods, results, and, more or less, in the discussion.
Otherwise, when the reader looks for the answer to: “we estimate the number and frequency of reptile relocation requests”
It becomes difficult to see how you answered that because you described all the details of the questionnaire without focusing on each of the questions you address.
“The second section quantified the extent of reptile management activity undertaken
by each participant.” Paragraph headings like this do not help find the relevant information hidden in the middle of the paragraphs.
Then in the results, you proceed as:
“3.2. Human-reptile interaction patterns”
Try to use the exact words instead.
For example:
In the methods section:
2.2. Number and frequency of reptile relocation requests regarding gender, age and income source.
Our questionnaire included the following relevant questions and factors….
In the results section:
3.2. Number and frequency of reptile relocation requests regarding gender, age and income source.
The high amount of sub-sections in the result section only find a lousy relation in the methods text, which indicates that you have left many objectives behind.
Then, in the discussion section, you can start the relevant paragraph similarly, or at least use the same order, which facilitates much for the reader.
I think you should apply that procedure to all the objectives. If you use these changes, you will see much improvement in the clarity and very likely polish some essential ideas in the text, possibly requiring more introduction/discussion to address all the actual objectives tackled in your manuscript and exhibited in the results section. The information is all there, and you write gracely; what your text still needs is the structure (and better resolution and formatting of the graphs, ouch).
Have present that the reader/reviewer has a busy mind and is trying to assess if what you say is supported by previous studies, by your data, or if it is just your interpretation. The suggested changes go towards facilitating that labor, which is paramount for solid science.
Best regards,
Agustín Camacho
Author Response
- The text has improved much, yet important concerns remain concerning its structure and graph formatting (the resolution is too low, and the text looks unformatted). As assistant editor of the journal, I praise the manuscripts for reaching an excellent writing level. Despite the additional work for everyone, it makes the article shine and makes potential future authors more willing to contribute to the journal and cite it, which benefits everyone.
Response 1: We thank the reviewer for taking the time to consider our manuscript so carefully and provide comments regarding the quality of our edits. We also thank the reviewer for a number of important suggestions and explanations for improvement. We have provided high resolution figures, however we are unsure what you mean by 'text looks unformatted' and happy for further clarification. We hope our thorough revision addresses the points raised and that our manuscript is now an acceptable contribution to the Diversity journal.
- The following hints should cover the problem.
Line 109. When you say we surveyed..or.we investigated, you enter in “methods” language. Keep on “Introduction” language and transform your objectives into questions or hypotheses you actually answered and/or tested in your methods. This is just fundamental for the reader to know what you did and assess if you did it right. Example (i) we estimate the number and frequency of reptile relocation requests, Transform to: “We asked/tested whether the number and frequency of reptile relocation requests would vary with the experience and gender of the operator and across different population segments.” See? In this way, you included the factors described in your methods and presented in your results. If you entered a test, state the hypothesis as part of your objectives. It is essential that the crucial variables tackled in each question have been introduced, and their obtention been described, reported and discussed. Otherwise, a lot of unuseful information blurs the important pieces composing the final message. Then, please remember these questions/objectives/hypotheses in your methods, results, and, more or less, in the discussion. Otherwise, when the reader looks for the answer to: “we estimate the number and frequency of reptile relocation requests”. It becomes difficult to see how you answered that because you described all the details of the questionnaire without focusing on each of the questions you address. “The second section quantified the extent of reptile management activity undertaken by each participant.” Paragraph headings like this do not help find the relevant information hidden in the middle of the paragraphs. Then in the results, you proceed as: “3.2. Human-reptile interaction patterns”. Try to use the exact words instead. For example: In the methods section: 2.2. Number and frequency of reptile relocation requests regarding gender, age and income source. Our questionnaire included the following relevant questions and factors….In the results section: 3.2. Number and frequency of reptile relocation requests regarding gender, age and income source. The high amount of sub-sections in the result section only find a lousy relation in the methods text, which indicates that you have left many objectives behind. Then, in the discussion section, you can start the relevant paragraph similarly, or at least use the same order, which facilitates much for the reader. I think you should apply that procedure to all the objectives. If you use these changes, you will see much improvement in the clarity and very likely polish some essential ideas in the text, possibly requiring more introduction/discussion to address all the actual objectives tackled in your manuscript and exhibited in the results section. The information is all there, and you write gracely; what your text still needs is the structure (and better resolution and formatting of the graphs, ouch). Have present that the reader/reviewer has a busy mind and is trying to assess if what you say is supported by previous studies, by your data, or if it is just your interpretation. The suggested changes go towards facilitating that labor, which is paramount for solid science.
Response 2: We thank the reviewer for their suggestions in breaking the sections under headings. We have incorporated each heading in the material and methods section, results section and used this format for the structure of the discussion. Each heading can now be found in the methods section on:
Line 145 - 2.1. Participant Demographics
Line 154 – 2.2. Number, frequency and taxa of reptile relocation requests regarding experience
Line 165-166 - 2.3. Decision-making process of reptile relocation requests regarding time of day, distance, and importance of environmental features for release
Line 179-180 - 2.4. Level of confidence relocated reptiles remain at the recipient site regarding experience and enterprise
Line 185 - 2.5. Number of injured wildlife and rehabilitation regarding experience and enterprise
Line 192 - 2.6. Number of escaped pets and exotic reptilian species regarding experience and enterprise
Line 201 - 2.7. Opinions about the relocation industry
We have also stated the questions we asked under each heading in the methods section:
i.e. Line 146 - 152 for 2.1. Participant Demographics
Line 155 - 163 for 2.2. Number, frequency and taxa of reptile relocation requests regarding experience
Line 167-177 for 2.3. Decision-making process of reptile relocation requests regarding time of day, distance, and importance of environmental features for release
Line 181-183 for 2.4. Level of confidence relocated reptiles remain at the recipient site regarding experience and enterprise
Line 186 - 190 for 2.5. Number of injured wildlife and rehabilitation regarding experience and enterprise
Line 193 – 199 for 2.6. Number of escaped pets and exotic reptilian species regarding experience and enterprise
Line 202 – 217 for 2.7. Opinions about the relocation industry
This manuscript is a resubmission of an earlier submission. The following is a list of the peer review reports and author responses from that submission.
Round 1
Reviewer 1 Report
Diversity report
Dear Editor, I have now read the article: “The reptile relocation industry in Australia: Perspectives from 2 Operators”. This article provides an interesting view on the issue of reptile relocation in Australia, raising many potentially important data. I only have one general concern, although severe: It arises from the fuzzy structure of the text and lack of important details in many of its segments. The flow of ideas is at times vague and hard to follow, the introduction should be reintegrated to become more directed to the point, which is to justify the questions answered in the article, which should also be stated as questions/hypotheses themselves, for clarity. Some segments are out of their proper section (i.e. results that should go in methods, discussion that should be results..), some results come from undescribed tests, and there are some conclusions that are not logically derived from the study. The objectives and methods are also not adequately reported. I have given many specific hints to the authors with the hope that they can adequate the text for a scientific manuscript. However, I think they should review more material on scientific writing before resubmitting, apart from following these hints.
Specific comments:
18-19: please be more concrete here.
20-21: please avoid repetition with results.
26-44: add species as examples and improve the flow of ideas. It reads too telegraphic now.
43: are they actually poor? Please add needed citation to support this affirmation.
57-59: take this part to demonstrate 43,
57: please avoid telling the reader (i.e. outcomes are not well studied), ans instead, keep “demonstrating” to the reader with facts from citations.
It seems that the examples in this whole paragraph should be integrated with the previous ones, making them much more concrete and fit for the present study.
The citations provided only show Australian anecdotal reports, yet there have been actually multiple translocation studies tracking located with non located individuals (ex. Hognose snakes in North America). Please provide these citations to increase the geographic coverage of the interest in this study and the understanding of the readers in the matter.
114-120. These are methods. Please declare the objectives shaped as questions about one variable (i.e. our objectives were to answer: How many reptile relocation requests receive Australian operators, monthly?), or hypotheses that relate two variables between them (i.e. we tested whether decision making in the process follows a previous protocol or not). These are just illustrative examples, no need to use precisely those.
142-144: Further describe the variables included in the survey (which are categorical (ex. paid or not), and how many categories they have, or then which are continuous (ex. salary amount). Which level of geographical area (municipal district, region?).
Likewise for all the variables described.
173: why not using the repetition of comments as an indicative of more widespread notions?
180-181: if this is a main result/test of the study, it should be introduced, stated as an objective and described precisely how the test was performed.
182: these are methods, take it to the relevant section.
193-194: same problem as in 180.
Figure 1.A. These categories should have been described in the methods section.
213: avoid imprecise statements “…just over a quarter of respondents...” , add the actual estimated frequencies (percentages) associated with each statement.
Fig.3: Appears laterally cut.
Figure 4. It would be relevant to compare these views of practitioners with what is known from studies empirically addressing these questions.
284: pythons are always non-venomous
283-296: very interesting data.
314-315: please add percentages for numbers. Also,a cross the results section, try to use the language to help the understanding of so many scattered data, by adding an easier to follow flow of ideas.
322-326: this should go in the methods section.
343: please associate numbers to percentages. And avoid imprecise statements as “some comments”.
Pasting so many comments transfers information very inefficiently, and feels boring. Why not conveying the most frequent messages, according to your analysis, and refer them to a graph or literal text in the supplementary material?
469: I did not see that being demonstrated. I did not even see an industry/commerce in their literal sense, please keep the wording precise and concrete.
492: avoid repeating results and expand them so readers can better understand them Which type of conflict?
This discussion section is extremely dry of previous studies, particularly those brought by actual studies on reptilian relocation across the world. Do not be afraid of losing novelty in your article and be proud of providing an advance and establishment of the understanding on the matter you are dealing with here.
501-523: good paragraph
535:538: Don’t tell, demonstrate. This conclusion is not logically derived from the previous statements.
549: which habitat use shifted? This paragraph finally compares views with knowledge. Please keep this pattern along the discussion, adding logically derived conclusions from such comparisons. If you go like that across all of the originally raised questions, we may get a nice piece of understanding on this purported “relocation industry”.
582-592: this segment is hard to understand, please restructure.
599-621: I would delete this paragraph or then relate it more to the question of this article, which is not about threats of invasive species but how relocation is done.
The conclusion is overlong and has not been constructed as it should, along the discussion. The conclusion should constitute a few lines that derive logically from all what has been demonstrated along the results/discussion.
Reviewer 2 Report
Dear Authors,
Overall, your manuscript is nicely presented. However, I have a few minor suggestions, mostly typographical.
i. the figures seem to have some formatting issues.
ii. line # 52 --> a space before the word "Additionally,"
iii. line # 69-70 --> change the font size
iv. line # 702 --> Pseudocheirus Peregrinus should be Pseudocheirus peregrinus
v. line # 726 --> please check the Symposium name
vi. line # 828 --> Trachemys Scripta Elegans should be Trachemys scripta elegans
Thanks.
Reviewer 3 Report
I believe that the article needs a very deep remodeling to be accepted. The introduction is well planned but forgets to comment on the aspects related to the carring capacity of the system or the competence... of the releases with authoctonous populations of reptiles. On the other hand, although it does appear in the discussion, it does not pose health problems for local species derived from exotic species.
It is not clear if pet includes feral cats.
The study suffers (lacks) from a more robust statistical treatment. Only chi-square results are presented, but a more complete analysis could have been attempted with the variables obtained in the surveys (regression, factorial analysis, correspondences,...). In addition, it is not specified in material & methods which statistical analyzes have been used to compare the data.
In general paragraphs with lots of percentages and numbers are hard to read and follow. I recommend lightening the text by replacing some of these paragraphs with pie or bar charts. (or table). It is not specific for one paragraph. Resuts is really hard to read.
Section 3.5 is not very relevant. I don't think they can be considered results. They are not categorized or analyzed. This section is simply a list of opinions without any type of analysis. Authors should try to categorize answers and create a table or graphs. Opinions are included in the Suppl. Mat. It is not needed repeat it in results. Categorize items and analyse it.
Some paragraph of 3.5 are methodology. (see comments in pdf)
An analysis of the published home ranges could have been carried out with the species that have data on release distances. It would be interesting but I don't know if this is possible.
Do you know if they are articles about carryng capacity of release habitats. It is an interesting criteria in order to determines release site.
Conclusions:
I think this section can be removed and the paragraphs incorporated into the discussion. Some ideas appear for the first time in the conclusions, so they could be considered part of the discussion.
Furthermore, I do not believe that new references should be included in the conclusions. It is more typical of the discussion. Remove any new references from the conclusions if the authors decide to keep the section. As currently written, they are not conclusions.
In my opinion the article in its current form cannot be accepted. You need to do a much more robust analysis of the data and not just present survey data as percentages. I think the article can be rejected or suggest major revision before resubmitting.
